# Myokines and Resistance Training: A Narrative Review

**DOI:** 10.3390/ijms23073501

**Published:** 2022-03-23

**Authors:** Beate E. M. Zunner, Nadine B. Wachsmuth, Max L. Eckstein, Lukas Scherl, Janis R. Schierbauer, Sandra Haupt, Christian Stumpf, Laura Reusch, Othmar Moser

**Affiliations:** 1Division of Exercise Physiology and Metabolism, University of Bayreuth, 95445 Bayreuth, Germany; beate.zunner@uni-bayreuth.de (B.E.M.Z.); nadine.wachsmuth@uni-bayreuth.de (N.B.W.); max.eckstein@uni-bayreuth.de (M.L.E.); lukas.scherl@uni-bayreuth.de (L.S.); janis.schierbauer@uni-bayreuth.de (J.R.S.); sandra.haupt@uni-bayreuth.de (S.H.); laura.reusch@uni-bayreuth.de (L.R.); 2Faculty of Medicine, Friedrich-Alexander-University Erlangen-Nürnberg (FAU), 91054 Erlangen, Germany; christian.stumpf@klinikum-bayreuth.de; 3Medical Clinic 2, University Hospital Erlangen, 91054 Erlangen, Germany; 4Heart Center Bayreuth, Department of Cardiology, Hospital Bayreuth, 95445 Bayreuth, Germany

**Keywords:** myokine, resistance training, Irisin, IL-6, myostatin, PGC-1 alpha, BDNF

## Abstract

In the last few years, the muscular system has gained attention due to the discovery of the muscle-secretome and its high potency for retaining or regaining health. These cytokines, described as myokines, released by the working muscle, are involved in anti-inflammatory, metabolic and immunological processes. These are able to influence human health in a positive way and are a target of research in metabolic diseases, cancer, neurological diseases, and other non-communicable diseases. Therefore, different types of exercise training were investigated in the last few years to find associations between exercise, myokines and their effects on human health. Particularly, resistance training turned out to be a powerful stimulus to enhance myokine release. As there are different types of resistance training, different myokines are stimulated, depending on the mode of training. This narrative review gives an overview about resistance training and how it can be utilized to stimulate myokine production in order to gain a certain health effect. Finally, the question of why resistance training is an important key regulator in human health will be discussed.

## 1. Introduction

Resistance training is known to be effective for maintaining and gaining muscle mass and strength [1]. Concerning mental-and physical health, this type of training stood in the shadow of endurance exercise, known for its positive effects on cardio-vascular health and functional capacity. This rethinking started to change with the discovery of myokines, a group of cytokines, small peptides, and proteoglycan peptides, secreted by contracting skeletal muscle cells [2]. Paracrine, autocrine and endocrine effects of these myokines were detected [2,3]. This led to a new understanding for the importance of muscular activity for human health. As discussed in this review, myokines have positive effects on metabolic, cardiovascular, mental, and immunological processes [4,5,6,7,8]. It is already well established that the human body possesses a complex muscle–body–crosstalk [3,8]. By stimulating the skeletal muscle in a certain way, we can make use of this cross talk and improve health [4,5,6,7,8].

Resistance training offers a variety of training methods to stimulate skeletal muscles that are more or less popular. For instance, resistance training with a goal of promoting muscle hypertrophy is triggering muscular growth by a stimulus to enhance muscle fibre growth leading to building muscles, so we can even enhance the health effects of our working muscle just by moving it in an appropriate way [9]. This narrative review will focus on the health-benefits of resistance training in general by explaining the “myokinological” background and enlighten the importance of resistance training in the therapy of the most common non-communicable diseases [5,10,11]. In Section 3, we will describe the investigated types of resistance training.

## 2. The Healthy Muscle Cell and Its Reaction to Resistance Training

Regular repetitive training leads to adjustment processes on neuronal and morphological levels. The neuronal adaptations lead to an increase of maximal strength and quick force in the early training phase. Following this first phase, morphological adaptations occur when regular training continues. These adaptations are mainly induced by three mechanisms: 1. mechanical tension, 2. muscle damage and 3. metabolic stress [12]. These mechanisms can lead to increased protein biosynthesis, enhanced endocrinological activity and activation of satellite cells (Figure 1). This in turn stimulates hypertrophy of myofibrils, changes in the muscle fibre spectrum and hypertrophy of sarcoplasm and connective tissue structures [12].

### 2.1. Mechanical Tension

Contraction and stretching generates mechanical tension in skeletal muscle, which seems to be the most important trigger leading to fibre hypertrophy by inducing molecular cascades that lead to protein synthesis; this process is called mechanotransduction [13]. The signaling involves increased levels of phosphatidic acid (PA) via diaglycerol kinase zeta (DKGz) [14]. PA in turn induces the mammalian target of rapamycin (mTOR), which is well known as the key regulator in protein synthesis [15]. Mainly responsible for this effect are tension dependent Ca^2+^ ion channels and membrane-bound mechanosensors like Integrine [16], and they react to the amount and duration of a mechanical stress [12].

### 2.2. Muscle Damage

Exercise induced microtrauma (muscle damage) is another cause for hypertrophy [12]. The extent of the damage depends on the intensity and duration of exercise, as well as the method of training and the training level of the athlete [12]. Microtrauma leads to an inflammatory reaction, which causes infiltration with granulocytes and macrophages. Cytokines and growth factors are released. This in turn activates the satellite cells [17]. Satellite cells belong to the pool muscle stem cells and are localized in the extracellular matrix, between sarcolemma and basal membrane [18]. After activation, these cells start to divide and migrate to the place of damage. Proliferating satellite cells are called myogenic precursor cells (MPC) [19]. Depending on the further steps of cell division, new myotubes or whole muscle fibres are created, either as single MPC, or as multiple melted MPCs [20]. While proliferating, these MPCs also create new satellite cells to refill the pool of muscle stem cells [19,20,21]. Under optimal conditions, the new muscle tissue is innervated, vascularized, and cannot be distinguished from uninjured muscle tissue, neither morphologically nor functionally [21].

### 2.3. Metabolic Stress

Examining the diverse metabolic changes in the muscle cell during resistance training shows that two further players are involved: the mTOR complex, also called Master Growth Regulator [15], and the AMPK- PGC-1 alpha axis—the energy regulators [22].

Once a contraction stimulus is received by muscle cells, ATP and Ca^2+^ are required for muscular action [23]. The available ATP lasts only for a few contractions. In the next step, the cell uses creatine phosphate to resynthesize ATP. This short-lived energy source is consumed within seconds. Then, the anaerobic glycolysis follows, producing not only ATP, but also lactate and NADH/H+ and phosphate (Pi) [24]. When these anaerobic metabolites accumulate, the pH level of the cell decreases. This change is relayed to the hypothalamus via chemosensitive afferent pathways where the secretion of growth hormones (GH), testosterone, as well as certain anabolic myokines, specifically IL-6, is induced. The described mechanisms are responsible for the induction of mTOR, the regulator of the protein synthesis [15]. Physical changes related to cell volume and hydrostatic pressure also occur, which after being recognized by certain mechanosensors, lead to the induction of mTOR [16].

If the contraction of the cell continues, the available energy resources quickly decrease, resulting in the rise of AMP concentration. AMP then activates AMP-kinase, which leads to enhanced energy uptake into the cell, activating lipid oxidation. AMP-kinase activates PGC-1 alpha by phosphorylation, therefore stimulating mitochondrial biogenesis [25]. In short, PGC-1 alpha plays a central role in the regulation of energy and mitochondrial biogenesis, as well as in oxidative stress defense mechanisms. A high activity of AMP-kinase inhibits the mTOR- pathway [12]. This means that when energy levels decline due to high energy demanding processes, the protein synthesis is stopped. Both pathways are in competition with each other [20] (Figure 1).

## 3. Different Types of Resistance Training

Concerning myokine production, there are mainly two investigated types of resistance training so far: hypertrophy training and strength endurance training. Further literature concerning other types of resistance training like maximal strength training is missing to date. The reason for the specific interest in hypertrophy and strength endurance training is the cellular signal cascade following the training stimulus, as mentioned above (Figure 1). The AMP-kinase pathway and the mTOR–pathway seem to be the key point in the production of certain important myokines [12,15,20,26].

In the following, we will give a short characterization of these two types of training.For a consistent definition of the investigated types of resistance training, we used the ACSM guidelines [27]. The specifications of repetitions and percent of the one repetition maximum are only examples. Due to the focus of this review on myokines, we renounce further training recommendations.

### 3.1. Hypertrophy Training

The current state of evidence suggests that the total volume of training is important to maximize muscle growth [28]; however, generalized loads cannot be prescribed due to the individual responses to resistance training [28]. This means that there are several recommendations concerning the definition of hypertrophy training [29]. The comparability of hypertrophy training study protocols is therefore impeded [29,30]. The definition for hypertrophy training and strength endurance training used in this review are taken from the American College of Sports Medicine position statement [27]: “An extensive utilization of energy-rich phosphates along with the consecutive accumulation of metabolites like H+, P (i), ADP, AMP and lactate is recommended. For example: 6–12 repetitions with 70–85% of the one repetition maximum (1 RM), in 1–3 series, with 1–3 min rest intervals between the series” [31] is recommended to attain this target. The rest period is required for the recovery of creatine phosphate, but not for a full metabolic recovery. The goal of hypertrophy training is to enhance the muscle fibre cross-sectional area (fCSA) in the slow Type-I, and mainly in the fast Type-IIA fibres, and to promote creatine phosphate and glycogen storage [1]. Furthermore, there is an improvement of the alactic as well as lactic metabolism via increased enzymatic activity [26].

### 3.2. Strength Endurance Training

In contrast to hypertrophy training, strength endurance training involves the use of lower weights, usually 50–70% of 1 RM, performing more repetitions [31]. For example, 1–3 sets with 10–15 repetitions for beginners, or multiple sets with 10–25 repetitions, or even more, for advanced athletes, rest intervals should last <90 s [27]. Extreme settings of strength endurance training are reported to work with 150 repetitions in less than 5 min per set [32].

The main goal of strength endurance training is not an increase of the fCSA, but an increase of time to exhaustion and maximal aerobic power [20]. These effects are unaffected by changes in VO_2_ *_max_* and seem to be related to increases in lactate threshold and lower limb strength, as well as improving running or cycling economy [33]. Just like hypertrophy training, strength endurance training has the potential to cause conversions from Type IIB to Type IIA muscle fibres [34].

The transitions from resistance training to strength endurance to muscle endurance is flowing [27]. There is no hard threshold; it seems more likely to be a continuum, a “strength-endurance continuum”, with high load/low repetitions on the one side and low load/high repetitions on the other side [32,33] (Figure 2).

## 4. Myokines

Myokines, which are secreted by muscle contractions, are able to restore metabolic balance in the human body and are able to regain health. Several hundred myokines have been identified to date, and presumably many hundred more are waiting to be discovered [35]. For simplicity, the review focuses on the most prominent myokines for potentially playing a role in the treatment of common non-communicable diseases such as the metabolic syndrome, cancer and neuronal degenerative diseases.

### 4.1. IL-6

Interleukin-6 (IL-6) was the first myokine that was discovered and is one of the most popular myokines discussed in several original data and reviews [6,7,11,36,37,38,39,40,41,42,43,44,45,46,47,48,49,50,51]. Years before the discovery of myokines, IL-6 was mentioned as an interferon (interferon beta2) [52]. In 1989, the molecule received its new name “Interleukin-6” [39]. In 2003, Pederson et al. presented this molecule as a potential “exercise factor” [7,41].

Although we know that IL-6 is a pro-inflammatory cytokine, it also has anti-inflammatory capacities when it is released by the working muscle [39]. The measurable plasma levels can increase up to 100-fold of the pre-exercise baseline and decrease back to baseline in a short period of time [53]. The anti-inflammatory effects seem to depend on the muscle as the secreting organ and a rapid onset peak effect. Furthermore, IL-6 seems to induce the production of other anti-inflammatory cytokines and to inhibit the production of TNF alpha and IL-1 beta [39,54]. Another important target for IL-6 is the metabolic system where it plays an important regulatory role by enhancing insulin-sensitivity and the accumulation of GLUT-4 [42]. IL-6 exerts its effect both locally within the muscle through activation of AMPK, and, when released into circulation, peripherally in a hormonelike fashion [39].

IL-6 not only has anti-inflammatory effects but is also involved in stimulating the multiplication of satellite cells in the muscular system, leading to hypertrophy of the muscle [55]. There are also hints that repetitive eccentric contraction leads to an increase in the number of stem cells and fusion of muscle fibres when compared against concentric training [56]. Via activation of the mTOR- signal cascade, IL-6 increases the protein synthesis in the myotubes [57].

Although IL-6 plays a very important role in exercise-metabolism, putting it in a pill seems not to be recommendable. High circulating levels of IL-6 are associated with inflammation. The structural differences between muscle cell IL-6 and other forms of IL-6 are unknown. Furthermore, there are different pathways following the activation of IL-6 receptor (IL-6R) that are responsible for either proinflammatory or anti-inflammatory effects of IL-6 [58,59]. There might be more cofactors involved as well, which could be responsible for the variant effects of IL-6.

### 4.2. Myostatin Group

#### 4.2.1. Myostatin

Myostatin is the only myokine that is reduced by exercise. It was identified in 1997 as a negative regulator of skeletal muscle growth [60]. The molecule belongs to the transforming growth factor beta superfamily (TGF-ß) [61]. The function of Myostatin is to limit muscle growth during embryogenetic development, but it is also expressed in adulthood [62]. The activated Myostatin has a high affinity to the Activin type 2 receptor (ACTRIIB) [63]. This activation triggers a catabolic cascade via SMAD1/SMAD2 and activates the forkhead box transcription factors family (FOXO1,2,3) [64]. Furthermore, mTOR and AMPK pathways are inactivated [64].

High levels of Myostatin inhibit the satellite cell proliferation and differentiation and block the action of muscle fiber protein accretion [65]. High levels are significantly associated with muscle wasting diseases, myopathy and sarcopenia [66]. Myostatin can also be expressed in cardiomyocytes, thereby associated with the development of heart failure [67]. A positive correlation between obesity, insulin-resistance and high levels of Myostatin has also been observed [65,68]. One of the most prescribed drugs, glucocorticoids, stimulate the expression of Myostatin that leads to iatrogenic muscle loss [69,70].

Myostatin plays a central role in the regulation of muscle growth and therefore also in the regulation of metabolism. In the past years, there have also been several pharmacological developments concerning suppression of Myostatin as a target for curing muscular dystrophies [69,71].

This is only one example of pharmacological development concerning the use of the Myostatin-system for therapeutic purposes in muscle dystrophies or secondary cachexia. After 20 years of Myostatin research, a successful transition from mouse-model to human-model is still missing. Too many cofactors that need to be considered are still prohibiting the implementation of myostatin-inhibition in therapy [71]. This makes physical activity and exercise induced Myostatin-inhibition even more important for therapeutic use.

#### 4.2.2. Decorin and Follistatin

The Myostatin–Antagonist Decorin was discovered in 1991 and later classified as a myokine [72]. It suppresses the Myostatin-effect by binding this myokine [73]. Another study confirmed that a Decorin-overexpression could enhance the proliferation of myoblasts and lead to the growth of myotubes [74]. Within muscle-body-crosstalk, Decorin inhibits angiogenesis and tumorgenesis [75].

Follistatin is another important antagonist of Myostatin, which cannot only bind Myostatin, but also blocks Myostatin receptors [76]. It was first discovered in follicular fluid and was found to be an FSH-suppressing protein [77]. It was classified as an antagonist of the TGF-beta- superfamily. Besides Myostatin, it can also bind other members of the TGF-beta family [76,78]. Concerning the muscular system, Follistatin can induce the proliferation of satellite cells, leading to protein synthesis [79,80]. Follistatin was also able to accelerate the healing of muscle injuries in mice, and reduced muscle fibrosis [81].

### 4.3. BDNF

The brain-derived neurotropic factor (BDNF) is produced by a wide spectrum of cells, including skeletal muscle cells, cardiac myocytes, smooth muscle cells, and cells in the liver and brain [82]. BDNF was first discovered in 1982 in the brain of pigs, long before it was identified as a myokine [83]. As a neurotrophine, BDNF is involved in the differentiation of neurons, synaptic plasticity and neurogenesis in the amygdala, prefrontal cortex and hippocampus [84,85,86]. It is furthermore a positive regulator of axonal and dendritic growth in the central nerval system (CNS) [87,88]. These functions promise potential for BDNF- associated therapies in the treatment of neuronal diseases like Alzheimer’s disease, Parkinson’s disease or Huntington’s Chorea. BDNF also has important influence on other tissues. It is engaged in the endogenous reparation of myocardial and skeletal muscle cells [89,90]. BDNF activity is also negatively correlated with insulin resistance, obesity and blood glucose levels [91,92]. Via the hypothalamic melatonin pathway, BDNF can regulate metabolism and may protect against obesity [93].

In the muscle itself, BDNF can be released by satellite cells in the case of injury and is involved in the regeneration of the damaged muscle [94,95,96,97]. Via the AMPK-pathway, BDNF enhances the lipid oxidation in the muscle cell [98], and improves glucose utilization [99]. Several studies show that circulating BDNF levels are elevated after resistance training, independent of the kind of resistance training [100]. Furthermore, the BDNF levels increased after training with regular repetition over several weeks [101]. Therefore, resistance training performed regularly can increase the positive effects of BDNF in the human body.

In summary, BDNF is a multifunctional protein with organ protective capabilities and therefore a very interesting molecule for therapeutic potential regarding the treatment of non-communicable diseases.

The myokine BDNF has only auto- and paracrine functions, as far as is currently known [98]. This means that BDNF produced by the muscle cell cannot be responsible for the elevated circulating levels seen after resistance training. The enhanced levels observed after training must be produced by other tissues, such as liver or adipose tissue [4]. In general, resistance training stimulates not only the muscle cell, but also different other organs to produce BDNF.

It is important to know that BDNF cannot pass the blood–brain barrier. This is one reason why it is difficult to develop BDNF-associated therapeutics for the treatment of neuronal disease [102]. In 2010, Linker et al. were successful in channeling BDNF through the blood–brain barrier by using T-cells as a transporter [103]. In the following years, different studies were performed to investigate the effect of exogenous BDNF in patients’ brains [104,105,106]. Although there were positive results in vitro as well as in mice-models concerning Multiple Sclerosis, the positive results in human patients with Parkinson’s disease failed to appear [102]. In conclusion, the positive effects of BDNF in the brain that appear by performing resistance training did not appear when BDNF was applied in the form of a medication—neither directly nor via genetic modifications [102].

### 4.4. The PGC-1 Alpha Group

#### 4.4.1. Irisin

Irisin is one of the most recent myokines that is described to date [107]. Biochemically, it is a PGC-1 alpha dependent molecule, cleaved of the transmembrane protein FNDC5. Irisin is proposed to be an exercise induced myokine, which is able to induce the browning of white adipose tissue [107]. The process of browning leads to increased thermogenesis and enhanced energy metabolism [108,109]. Lively discussions regarding this molecule ensued, due to controversial opinions regarding the expression of Irisin. It was labelled as a mythos rather than a myokine [110]. This debate could be solved by using a different detection method. The formerly used commercial ELISA-kits seem to be rather unspecific: They detected not only Irisin, but also different cross-reacting proteins. Through implementation of tandem mass spectrometry, the results started to become clearer [111]. Another point that leads to confusion was the time of measurement. Since Irisin is a short-lived molecule, different times of measuring after training lead to different conclusions about the exercise induced expression of Irisin [112,113,114]. For later studies, this was considered. In the following years, numerous studies concerning Irisin were performed, underpinning Irisin’s existence (Table 1).

Irisin can be produced in muscle, adipose tissue and in the myocardium [115]. A powerful stimulator for muscle cells to produce Irisin is physical exercise. Among different exercise studies, resistance training crystallized to be most effective; however, this has not been confirmed until today [113,116,117].

The most interesting effect of Irisin is the browning of white adipose tissue by increasing the expression of mitochondrial uncoupling protein 1 (UCP 1) [107]. This leads to an activation of non-shivering thermogenesis. Furthermore, Irisin improves glucose homeostasis and lipid metabolism, and reduces insulin-resistance as well as adipose tissue inflammation [108,109,118]. An important benefit for the myocardium is the reduction of endothelial function abnormalities. Irisin protects the myocardium against ischemia and reperfusion injuries through the AMPK-Akt-eNOS-NO-pathway [119,120].

Additionally, Irisin seems to enhance bone mass with positive effects on cortical mineral density and bone geometry via alphaV/beta5-integrin [121,122].

Concerning neurological diseases, Irisin has positive effects on Alzheimer’s disease by reducing insulin-resistance and improving glucose metabolism. Irisin is also involved in the process of neurogenesis in the CNS [123].

In the case of cancer, there are varying results, but most results confirm a positive effect on oncologic diseases [124]. Studies show that Irisin induces cancer cell apoptosis and reduces the migration of breast cancer cells [125]. Irisin is also able to enhance the sensibility of cancer cells for Doxorubicin, a well-known chemotherapy agent. This led to the hypothesis that, in combination with Irisin, lower Doxorubicin doses could be sufficient [125]. Breast cancer patients also seem to have a lower base level of Irisin compared to a healthy population [126]. Similar positive effects could be found by treating colorectal-, prostate, lung-, bone-, and pancreas cancer cells [127,128,129,130]. Different results were found by investigating Irisin and hepatocellular cancer. An overexpression of Irisin mRNA was observed in liver cells [131], along with reduced sensibility for Doxorubicin [132]. Concerning cancer and Irisin, a general recommendation cannot be given, as multiple factors are involved—the primary factor is the type of cancer being studied. More studies are needed.

Though Irisin was first discovered in 2012, there are already several well-known drugs that influence PGC-1 alpha and Irisin. Metformin, a drug for treating diabetes type 2, enhances the expression of FNDC5 mRNA and circulating Irisin levels in mice [133]. These results have not yet been confirmed in humans. Other drugs for treating cardiac diseases (for example, Losartan, Catecholamine, Bezafibrate, Fenofibrate and Pioglitazone) are shown to stimulate PGC-1 alpha expression and activate a certain pathway [134,135,136,137,138]. Further studies are needed to investigate a potential correlation with Irisin.

Another point to consider is the dependence of Irisin on PGC-1 alpha. It is a central player involved with the regulation of mitochondrial biogenesis and cell metabolism and can be induced by various stimuli. This in mind could help identify heretofore unexplored ways of Irisin induction. Some of these have already been observed, like inducing Irisin through cold-exposure, or other external stimuli, leading to an increased ATP demand [139].

#### 4.4.2. Meteorin-Like

Meteorin-like (Metrnl) is another recently discovered adipomyokine, dependent on PGC-1 alpha. Metrnl was discovered in 2014 by using the splice variant PGC-1 alpha4 [140]. It was first found to be preferentially induced in muscle cells by resistance training and linked to muscle hypertrophy [141]. Later, another study showed that Metrnl is generally inducible by exercise. However, this study could not specify whether resistance training or endurance training is more likely to induce this myokine [37]. More studies are required.

Like Irisin, Metrnl is able to induce the browning of white adipose tissue, not by direct action on adipocytes but rather via recruitment of IL-4 and IL-13 expressing eosinophils in adipose tissue [140]. This leads to enhanced glucose tolerance and improved insulin sensitivity in mice. Even a loss of 25% body fat in the Metrnl treated mice could be observed [140]. These are reasons why Metrnl could be a potential target fighting metabolic diseases and obesity. Studies with humans were more complicated, as various cofactors (like patients’ medications) had to be considered [142]. This leads to different outcomes concerning Metrnl and diabetes type 2 patients [142]. Further studies need to be performed. Recently, Metrnl has been reported as a cytokine expressed by macrophages and barrier tissues like skin and respiratory tract epithelium [142,143]. It seems closely involved in innate and acquired immune responses and inflammatory regulation [144].

## 5. Myokines and Resistance Exercise

### 5.1. IL-6

Concerning resistance training, there are controversial results in different studies regarding measured post exercise IL-6 levels [45,145]. There might be a necessary minimum total volume of training [146], just like a threshold or a sweet spot, required to attain this peak effect of IL-6. The studies with the highest IL-6 output involved a full body resistance work-out including large muscle groups [44,146]. This hypothesis matches the described reaction of a muscle cell to an adequate stimulus, described in the first part of this review.

By taking a closer look on the studies, the biggest differences that occur are concerning the method of measurement of IL-6. Buford et al. used muscle biopsies for the purpose of determining IL-6-mRNA levels [145]. Della Gatta et al. also performed muscle biopsies, but for determining IL-6 itself and not IL-6-mRNA levels [147]. Mendham et al. and Phillips et al. measured serum-IL-6 levels after exercise [44,146]. Quiles et al. determined plasma-IL-6 levels after exercise [148].

All studies concerning the acute response after exercise could present an increase in their measured IL-6 levels, independent from the type of measurement (Table 1).

Tomeleri et al. used the baseline serum-IL-6 level as a main target [149]. This was the only human study which could be found, concerning resistance training and the baseline serum-IL-6 level. The decrease could be interpreted as an improvement of the inflammatory level.

### 5.2. Myostatin

As has already been mentioned, Myostatin operates as an inhibitor of muscle growth [60]. Therefore, lowering the Myostatin-level via training is the worthwhile goal for muscle growth [65]. Kazemi et al. were able to show that even a single session of exercise could reduce the plasma-Myostatin level [150]. Raue et al. found similar results: after one session of resistance exercise, the Myostatin-mRNA levels in the collected muscle biopsies were significantly reduced [151]. This means, for practical implementation, every session of resistance training counts.

### 5.3. Decorin

Decorin is an important antagonist of Myostatin [72]. This means that Decorin and Myostatin levels correlate inversely. The fact that resistance exercise is an appropriate method to enhance Plasma-Decorin levels, which could be demonstrated by Bugera et al. and Kanzleiter et al. [152,153]. Both studies performed their measurements after a single bout of exercise.

### 5.4. Follistatin

Different studies present the same results concerning the increase of plasma or serum-Follistatin levels after resistance training [154,155,156,157]. Bagheri et al. performed a randomized controlled trial over eight weeks, with the result of elevated serum-Follistatin, depending on the volume of activated muscle groups [154]. Increased serum-Follistatin was also the result in the study of Hofmann et al. in the Vienna Active Ageing Study [157]. Negaresh et al. could show an increase in plasma-Follistatin levels as well, independent from the age of the investigated volunteers [155].

### 5.5. Irisin

As described above, Irisin is a frequently discussed myokine. One of the reasons is the fact that we still do not exactly know how this molecule is stimulated in the most effective way. In Table 1, you can see the current number of studies performed to find a correlation between a certain type of exercise and Irisin production, with very different results.

Blizzard Leblanc et al. first compared a single bout of aerobic exercise with a single bout of resistance exercise, followed then by six weeks of resistance exercise (RE) [9]. The plasma-Irisin level did not change after a single session of RE, but after a single session of aerobic exercise (AE). After six weeks of RE though, there was a measurable increase in the plasma-Irisin level [9]. In contrast, Ellefsen et al. did not find any changes in the serum-Irisin level after 12 weeks of full body heavy strength training [158].

Huh et al. compared RE and AE. They found an elevated serum-Irisin in both groups, immediately after exercise, which returned to baseline 1 h after the end of training [117]. Another study which compared RE and AE was performed by Kim et al. in a randomized controlled trial [159]. In this study, only the RE group could present an elevated plasma-Irisin level. Norheim et al. performed a mixed form of training for the duration of 12 weeks and differed between inactive and prediabetic volunteers [160]. They also observed elevated plasma-Irisin levels with a stronger increase for the prediabetic group. Interestingly, the baseline level of Irisin was reduced after 12 weeks of training. Nygaard et al. compared AE and RE in a single session randomized cross-over design [161]. Once again, plasma-Irisin levels increased after both exercise interventions, but remained higher after RE. Pekkala et al. compared four different types of exercise protocols for a duration of 21 weeks, but they could not measure any change in the serum-Irisin levels [162].

Tibana et al. is one of the rare studies which did not show an increase but a decrease in plasma-Irisin levels of non-obese volunteers after 16 weeks of RE [163].

Tsuchiya et al. performed another randomized cross-over study with healthy men [116]. Plasma-Irisin levels increased in all three intervention groups, with a significantly higher increase in the RE group. Zhao et al. performed their study with core and leg strength training, and they also could find increased Serum–Irisin levels [164].

Although there are contradictory results now and then, the vast number of studies predicts a positive effect of RE concerning Irisin. There also seems to be an advantage over AE [116,161,165].

### 5.6. BDNF

For this review, 16 studies concerning BDNF and RE could be found, which fits the inclusion criteria mentioned above. Compared with the other myokines of this review, this is the largest number of studies. The strong interest in BDNF can be explained by its positive effects on neuronal diseases as already explained in the first part of this review using a resistance training protocol targeting strength (e.g., 5 × 85% 1 RM).

Church et al., Domínguez-Sanchéz et al. and Figueiredo et al. could all find an increase in plasma-BDNF levels after RE [89,101,166]. The study of Domínguez-Sanchéz et al. was performed as a randomized controlled trial with RE, high intensity interval training (HIIT) and a combined group [166]. The combined group showed the highest increase in serum-BDNF levels. Forte et al. also compared three groups of different types of RE [167]. Only in one group could an increase of serum-BDNF be detected.

Lodo et al. and Jørgensen et al. are two studies that could not find any change in serum- or plasma-BDNF levels after RE [168,169].

Marston et al. published two studies with different intervention protocols [170,171]. Both studies could show an increase in serum-BDNF, but only for the high load group in the study from 2019 [171], and a higher increase for the 3 × 10 repetition group in the study from 2017 [170]. This underlines the hypothesis of a minimum total load of exercise needed to gain a measurable effect.

McKay et al. performed muscle biopsies after a single bout of 300 × maximal eccentric contractions [90]. After this muscle damaging exercise, BDNF was elevated in the biopsies, and a correlation between BDNF and the proliferation as well as differentiation of satellite cells could be demonstrated for the first time [90].

Roh et al. and Urzi et al. both performed studies with elastic band resistance training in elderly women [172,173]. Both studies could detect an increase of the serum- or plasma-BDNF levels.

The study of Rojas Vega et al. could only find a non-significant increase of serum-BDNF after isokinetic knee extension [174]. A potential explanation for this result could be the upper mentioned minimum load hypothesis.

A huge study with several publications was the HEARTY-TRIAL [175,176,177,178]. Concerning BDNF and RE, Walsh et al. and Alberga et al. could demonstrate that the baseline serum BDNF level of obese adolescents was elevated after 22 weeks of training, either RE or AE, or combined [175,178]. In another study with older adults, Walsh et al. found elevated serum-BDNF levels after eight weeks of RE as well [179].

Wens et al. compared healthy persons with multiple sclerosis (MS) patients [180]. A lower baseline serum-BDNF level was noticed in the MS group. In both groups, serum-BDNF increased after RE.

Yarrow et al. found a higher response of serum-BDNF to RE at the end of a 5-week RE intervention, but no baseline change [100].

The wide range of studies performed with resistance training in order to elevate BDNF levels is impressive. Most of these studies could present positive results. This underlines the importance of resistance training in therapy and prevention of neuronal diseases.

**Table 1 ijms-23-03501-t001:** Summary of human studies investigating myokine secretion via resistance training protocols.

Myokine	Study	*n*	Study Cohort	Training Protocol	Trial	Results
**IL-6**	Buford et al. [145]	24	Physically active postmenopausal women	3 × (10 × 80% 1 RM)	1 session	Muscle biopsies: IL-6-mRNA ↑
	Della Gatta et al. [147]	16	Young men *n* = 8; elderly men *n* = 8	2 × (8–12 × 50–80% 1 RM), progressive increase	12 weeks	Muscle biopsies: IL-6 ↑; not significant different between the groups
	Mendham et al. [44]	12	Sedentary men	3 × (10 × 60% 1 RM) vs. 3 × (10 × 80% 1 RM) vs. Low intensity aerobic exercise (40 min) vs. Moderate intensity aerobic exercise (40 min)	4 sessions, randomized cross-over	Serum- IL-6 ↑ in the moderate intensity groups (RE and AE)
	Phillips et al. [146]	14	Healthy men	3 × (12 × 65% 1 RM) vs. 3 × (8 × 85% 1 RM)	2 sessions, controlled	Serum-IL-6 ↑, with association to total volume load
	Quiles et al. [148]	15	RE-experienced men	4–5 × (8–12 × 60–70% 1 RM) vs. 8–10 × (2–6 × 75–85% 1 RM)	3 weekly, 6 weeks, randomized	Plasma-IL-6 ↑ in both groups; no significant effect on BDNF
	Tomeleri et al. [149]	38	Obese older women	3 × (10–15 maximum repetitions)	3 × weekly, 8 weeks, randomized controlled	Baseline-Serum-IL-6 ↓
**Myostatin**	Kazemi et al. [150]	24	Healthy men	3 × (15 × 55% 1 RM)	1 session	Plasma-Myostatin ↓
	Raue et al. [151]	14	Young women *n* = 8 Old women *n* = 6	3 × (10 × 70% 1 RM)	1 session	Muscle biopsies: Myostatin mRNA ↓
**Decorin**	Bugera et al. [152]	10	Physically active young men	4 × (7 × 80% 1 RM) vs. 4 × (15–30 × 30% 1 RM) vs. BFR 3 × (15–30 × 30% 1 RM)	1 session	Plasma-Decorin ↑
	Kanzleiter et al. [153]	10	Young men	3 × (8 × max weight)	Human study I, single bout	Plasma-Decorin ↑
**Follistatin**	Bagheri et al. [154]	40	Middle aged men	3–4 × (15 > 12 > 10 > 8 × 50–80% 1 RM) Upper body vs. lower body vs. both vs. control	3 × weekly, 8 weeks, 10% increase every 2 weeks, randomized controlled	Serum-Follistatin ↑, serum-Myostatin ↓, depending on the volume of activated muscle mass
	Hofmann et al. [157]	91	Elderly women	1–2 × 15, elastic bands vs. Training + nutritional supplements vs. Cognitive training	Progressive increase of resistance; 6 months; randomized	Serum-Follistatin ↑ only in the training group
	Negaresh et al. [155]	31	Elderly men *n* = 15 Young men *n* = 16	4 × (10 × 50%−85% 1 RM)	3 × weekly, 8 weeks, 5% increase per week	Plasma-Follistatin ↑; plasma-Myostatin ↓ in both groups
**Irisin**	Blizzard Leblanc et al. [9]	11	Obese youth	4 × (12–15 × 60–65% 1 RM)	3 × weekly, 6 weeks	Plasma-Irisin → after a single bout of RE, but ↑ after a single bout of AE; with greater ↑↑ after 6 weeks of RE
	Ellefsen et al. [158]	18	Untrained women	Progressive full body heavy strength	3 × weekly, 12 weeks	Serum-Irisin →
	Huh et al. [117]	20	Sedentary healthy men *n* = 14 Men with metabolic syndrome *n* = 6	3 × (8–12 × 75–80% 1 RM) vs. HIIT (4 ×4 min 90% Vo2 max) vs. Continuous moderate exercise (36 min 65% Vo2 max)	1 session, randomized cross-over	Serum-Irisin ↑ immediately and ↓ after 1 h back to baseline; no difference between the groups
	Kim et al. [159]	28	Obese adults	3 × (10–12 × 65–80% 1 RM) vs. Aerobic exercise (50 min 65–80% HR max)	5 × weekly, 8 weeks, randomized controlled	Plasma-Irisin ↑ only in the RE group
	Norheim et al. [160]	26	Healthy physically inactive men *n* = 13 Pre-diabetic men *n* = 13	Combined strength and endurance training, 2 × weekly 60 min ergometer and 2 × weekly 60 min full body strength workout	4 × weekly, 12 weeks	Plasma-Irisin higher in the prediabetic group; acute ↑ after exercise; baseline ↓ after 12 weeks
	Nygaard et al. [161]	9	Moderately trained, healthy adults	3 × (10–12 × max weight) vs. 6 × 5 min high intensity treadmill, Borg > 18	1 session, randomized cross-over	Plasma-Irisin ↑ in both groups, but remained higher in the RE group
	Pekkala et al. [162]	56	Untrained healthy men	Aerobic exercise (60 min 50% Vo2 max) vs. 5 × 10 repetitions until failure (leg press only) vs. Long term endurance exercise vs. Long term RE and endurance exercise	Single bout vs. 2 × weekly, 21 weeks	Serum-Irisin →
	Tibana et al. [163]	49	Inactive women: Obese *n* = 26 Non obese *n* = 23	3 × (6–12 × repetitions maximum)	2 × weekly, 16 weeks	Plasma-Irisin ↓ in the non-obese group after intervention; no change in the obese group
	Tsuchiya et al. [116]	10	Healthy men	3–4 × (12 ×65% 1 RM) vs. Aerobic exercise (60 min 65% Vo2 max) vs. 30 min RE + 30 min AE	3 single bouts of exercise; randomized cross-over	Plasma-Irisin ↑, significant higher in the RE group
	Zhao et al. [164]	17	Older male adults	Class of leg muscle strength and core strength training	2 × weekly, 12 weeks, randomized controlled	Serum-Irisin ↑
**BDNF**	Church et al. [101]	20	RE-experienced young men	4 × (10–12 × 70% 1 RM) vs. 4 × (3–5 × 90% 1 RM)	4 × weekly, 7 weeks	Plasma-BDNF ↑
	Domínguez-Sanchéz et al. [166]	51	Physically inactive, obese men	(12–15 × 50–70% 1 RM) vs. HIIT vs. HIIT + RE	1 session, randomized controlled	Plasma-BDNF ↑, highest ↑↑ in the combined group
	Figueiredo et al. [89]	21	Physically active men	1 min 100%VO2 max + 8 exercises 8–12 RM	8 weeks, control group	Plasma-BDNF ↑
	Forti et al. [167]	65	Healthy elderly	2 × (10–15 × 80% 1 RM) vs. 1 × (80–100 × 20% 1 RM) vs. 1 × (60 × 20% 1 RM) + 1 × (10–20 × 40% 1 RM)	3 × weekly, 12 weeks, randomized	Serum-BDNF ↑ in male participants, only in the 3rd group
	Lodo et al. [168]	20	Young healthy adolescents	4 × (5 × 70% 1 RM) vs. 4 × (10 × 35% 1 RM)	2 bouts of exercise with equated total load lifted	Serum-BDNF →
	Jørgensen et al. [169]	30	Persons with multiple sclerosis	Progressive high intensity	2 × weekly, 24 weeks, randomized controlled	Plasma-BDNF →
	Marston et al. [171]	45	Healthy adults	5 × (5 × 85% 1 RM) vs. 3 × (10 × 70% 1 RM) vs.	2 × weekly, 12 weeks, randomized controlled	Serum-BDNF (↑) only in the high load group
	Marston et al. [170]	16	Untrained men *n* = 11 Untrained women *n* = 5	5 × 5 repetitions to-fatigue vs. 3 × 10 repetitions to fatigue	2 bouts of exercise; cross-over	Greater serum-BDNF ↑ in the second group
	McKay et al. [90]	29	Male adolescents	300 × maximal eccentric contractions	Single bout	Muscle-biopsy: BDNF ↑
	Roh et al. [172]	26	Elderly, obese women	Elastic bands, intensity:10–14 RPE	3 × weekly, 12 weeks, randomized controlled	Serum-BDNF ↑
	Rojas Vega et al. [174]	11	Healthy adults	3 repetitions of maximal effort isokinetic work (knee extension): 40% 1 RM vs. 110% 1 RM	2 single bouts	Serum-BDNF (↑), but no significance
	Urzi et al. [173]	20	Elderly women	Elastic band resistance training	12 weeks, randomized controlled	Plasma-BDNF ↑
	Walsh et al. [175] Alberga et al. [177,178]	202	Postpubertal adolescent with obesity	HEARTY-Trial: 2–3 × (6–15 × maximum reps) vs. 20–40 min 70–85% HRmax vs. combination	4 × weekly, 22 weeks, randomized controlled	Baseline plasma-BDNF ↑
	Walsh et al. [179]	10	Older adults	4 × (8–10 × 60–80% 1 RM)	3 × weekly, 8 weeks	Serum -BDNF ↑
	Wens et al. [180]	41	Persons with multiple sclerosis *n* = 22 Healthy persons *n* = 19	1 × (10 × 12–14 RPE) increased to 4 × (15 x12–14 RPE)	5 sessions per 2 weeks, 24 weeks, randomized controlled	Baseline serum-BDNF were lower in persons with MS; Serum-BDNF ↑ after exercise in both groups
	Yarrow et al. [100]	20	Healthy young adults	4 × (6 × 52,5% 1 RM) trad. vs. 4 × (6 × 40% 1 RM) concentric vs. 3 × (6 × 100% 1 RM) eccentric	5 weeks	Serum BDNF ↑, with higher response at the end of intervention. No baseline change

Abbrevations in order of appearance: RM: repetition maximum; RE: resistance exercise; AE: aerobic exercise; HIIT: high intensity interval training; RPE: rate of perceived exertion; MS: multiple sclerosis.

## 6. Discussion—Myokines and Resistance Training

This review shows that resistance training can induce the production of different myokines, which are meaningful to retain or even regain health. The anti-inflammatory effects (IL-6) [39], the enhancement of insulin sensitivity (Mtrnl) [140] or the browning of adipose tissue (Irisin, Mtrnl) [107,140] with subsequent optimizing of the status of metabolic diseases, and many other positive effects, as highlighted in this review, underline the importance of giving resistance training the status of a medical treatment for certain diseases [181]. The myokines IL-6 and Irisin seem to develop their positive effects directly by inducing certain pathways [54,107], whereas Mtrnl first recruits eosinophils in adipose tissue [140]. Follistatin and Decorin develop their effects by being an antagonist of Myostatin and binding this myokine [73,76]. BDNF seems to be produced by a variety of tissues, such as muscle, liver, adipose tissue and brain [82].

Depending on the therapeutical goal, you can activate either the mTOR- pathway, or the AMPK-pathway for myokine production (Figure 1 and Figure 2). It seems obvious that metabolic diseases like Type 2 Diabetes or obesity and cardiovascular diseases could benefit from the activation of the AMPK-pathway. Most studies considered in this review just focused on myokine production itself, if myokines can be stimulated via resistance training at all. This is a first step and is important to know, but now we have to look further. We have to consider body composition, gender, age, and probably even hormonal status as well for future studies.

We could also show that there are various types of exercise protocols, investigated to their beneficial effect on myokine production. The studies included in this review are very heterogeneous. Population, study designs and even resistance training criteria are not comparable to each other. However, most studies have one thing in common: they induce myokine production.

Some of the Myokines, like Irisin, seem to be more stimulated via resistance training compared to endurance training. Another point of interest seems to be the intensity of exercise. Therefore, different types of resistance training were investigated in the considered studies for this review. Hypertrophy training stimulated different types of myokines compared to strength endurance training, although the transition between both is flowing (Figure 2). Most protocols used the method of “1 Repetition maximum” (1RM), or other variants of RM, to find the required training zones. This tool is an important factor for the comparability of the studies. Investigators therefore have the opportunity to use a certain protocol for different groups of participants, compare the results and adapt the training to the possibilities of each individual participant. We recommend for further investigations to use the ASCM-guidelines [27] to give a clear definition of the investigated types of training, to make further studies more comparable.

However, as already considered above, the myokine stimulation seems to depend on more factors than only on the intensity and duration of training including cofactors like nutrition, lifestyle, medication, circadian rhythmic and several more influences, presumably the myokine production [182]. (Figure 3) Therefore, we should consider a multidimensional context in the exercise induced cell metabolism for further studies. Particularly, we should pay attention to the nutritional influence on cell metabolism. We already know, for example, that amino acids can induce mTOR and lead to protein biosynthesis [183]. Some studies already investigate the influence of nutritional facts on myokine production [182].

## 7. Conclusions

The difficulty of interpreting the results of these studies is the wide variety of exercise protocols and participants. A “one fits all” recommendation concerning the right training protocol for a certain person, regarding health and training status, for the stimulation of a certain myokine, cannot be given at present. This fact is in turn not only the result of this wide range, but it is also the reason why so many different study and exercise protocols exist. However, we can see a direction, and we now know that resistance training is a crucial part of maintaining health [184,185]. More studies are needed to investigate individual cofactors like nutrition, illness, and lifestyle, to specify individual training recommendations.

We already know a lot about the healthy muscle metabolism [186], but we need to know more about the different types of disruption of muscle metabolism, with the intention of creating a healing individual training protocol for every type of patient.

## Figures and Tables

**Figure 1 ijms-23-03501-f001:**
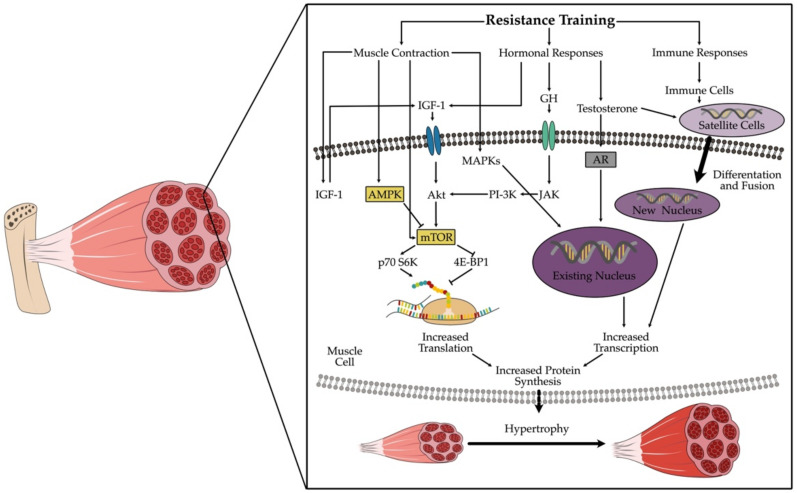
Intracellular adaptions following resistance training and leading to increased protein synthesis. Only the most important steps are pictured, the processes in vivo are much more complex. Abbreviations: Akt = protein kinase B; AMPK = adenosine monophosphate-activated protein kinase; AR = androgen receptor; GH = growth hormone; IGF-1 = insulin-like growth factor 1; JAK = janus kinase; MAPKs = mitogen-activated protein kinase; mTOR = mammalian target of rapamycin; PI-3K = phosphatidylinositol-3 kinase; p70 S6K = 70 kDa ribosomal protein S6 kinase; 4E-BP1 = eukaryotic initiation factor 4 E.

**Figure 2 ijms-23-03501-f002:**
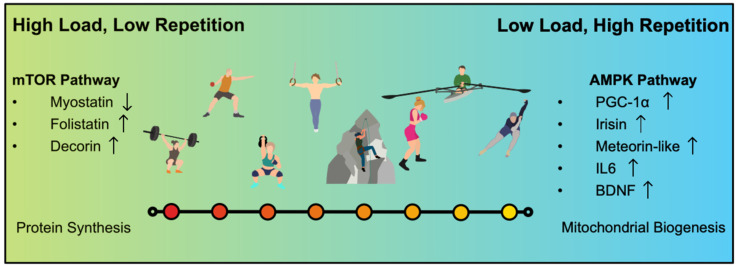
The Strength–Endurance Continuum shows exemplarily different types of sports, associated with strength training. In general, there is not only weightlifting, but also other kinds of strength training that lead to the health promoting effects.

**Figure 3 ijms-23-03501-f003:**
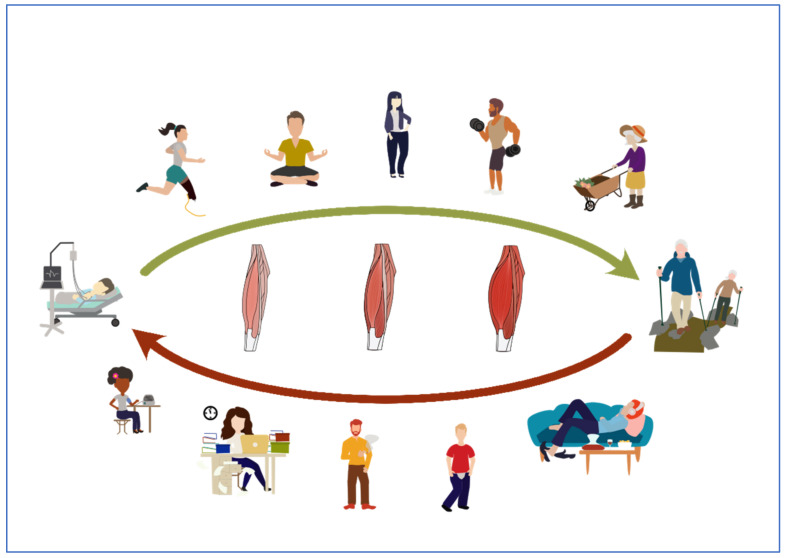
Potential factors that influence the myokine production. Not only sports, but also body weight management, stress reduction, nutrition, daily movement, and other lifestyle factors could enhance the myokine production. Vice versa, an unhealthy lifestyle could lead to loss of muscle mass and an impaired myokine production.

## Data Availability

Not applicable.

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
