# Peer review of "Myokines and Resistance Training: A Narrative Review"

_ijms, 2022, doi:10.3390/ijms23073501_

Round 1

Reviewer 1 Report

The authors review recent literature on resistance training and signaling molecules released by muscles, called myokines. The review is extensive, and the authors describe and review the literature on numerous cytokines. While this provides a comprehensive review, that will be a good resource, the overall synthesis of the data can be improved. I would suggest that the authors add another paragraph at the end summarizing and adding some color to the manuscript that would allow the reader to pickup on treads and future directions in the field.

Overall, the manuscript is well written and easy to read. The strengths and weakness of each published study are well described and this provides a very helpful resource to the field. The only critique is that the itemized accounting of the literature approach needs to be complemented with some synthesis or "sum of the parts" view, perhaps at the end of the manuscript.  

Reviewer 2 Report

With the review "Myokines and resistance training: a narrative review", authors aimed to give an overview about the resistance training impact on health through the myokines acitivity. The review is well presented and figures and table are useful for an easy understanding of the summarized concepts. The topic is interesting but some important aspects should be considered.

Why authors chose human studies based on criteria not comparable to each other? I suggest to insert or to divide in categories the studies on the base of criteria (i.e disease, treatments, type of trainig, gender...)

I'd like to ask to authors which is the contribution of inflammation in the myokines-mediated exercise training? Are there evidence about the mechanisms by which myokines can exert beneficial effect on health?

Which parameters were measured to assess the health benefits? 

Type of resistance training should be explained in the introduction.

Also, I woul add a method section explaining the criteria selection of the studies, the evidence recruitment and the bibliography selection.

See the PRISMA2020 guidelines: Page, M.J., McKenzie, J.E., Bossuyt, P.M. et al. The PRISMA 2020 statement: an updated guideline for reporting systematic reviews. Syst Rev 10, 89 (2021). https://doi.org/10.1186/s13643-021-01626-4

Minor:

Correct IL6 in IL-6 in discussion section

Correct uppercase or lowercase in the names of myokines

Discussion is very approximate. 

Reviewer 3 Report

The authors make a   narrative review  focused on the health-benefits of resistance training in general by explaining the myokinological background and enlighten the importance of resistance training in the therapy of the most common non communicable diseases .

The authors carry out a comprehensive review, with associated iconography, trying to explain the benefits of myokines in health and disease situations, as well as the effects of different types of training. The review is well written, up-to-date, with significant bibliographic support.

Round 2

Reviewer 2 Report

Authors responded satisfactorily to some points that deserved consideration.